# Integrated Identification and Genetic Diversity of Potentially Invasive Clearwing Moths (Lepidoptera: Cossoidea: Sesiidae) in Korea

**DOI:** 10.3390/insects15010079

**Published:** 2024-01-22

**Authors:** Sora Kim, Jong-Kook Jung, Ikju Park, Bong-Woo Lee, Yong-Hun Kim

**Affiliations:** 1Laboratory of Insect Phylogenetics and Evolution, Department of Plant Protection & Quarantine, Jeonbuk National University, Jeonju 54896, Republic of Korea; 2Department of Agricultural Convergence Technology, Jeonbuk National University, Jeonju 54896, Republic of Korea; 3Department of Forest Environment Protection, Kangwon National University, Chuncheon 24341, Republic of Korea; jkjung@kangwon.ac.kr; 4Department of Entomology, University of California Riverside, Riverside, CA 92521, USA; ikju.park@ucr.edu; 5Gwangneung Forest Conservation Center, Korea National Arboretum, Pochoen 11186, Republic of Korea; eucosma@korea.kr; 6Forest Pests & Diseases Control Division, Korea Forestry Promotion Institute, Dajeon 35209, Republic of Korea; jaabba@kofpi.or.kr

**Keywords:** invasive insect pests, regional populations, genetic divergence, COI, street insect pests

## Abstract

**Simple Summary:**

Climate change is accelerating the introduction of exotic pests, coupled with an increase in international trade and travelers. This study is a case of a street tree planted as an urban green space for natural landscape, shade, and pollution prevention that became a habitat for exotic pests. As an initial response to the exotic pests damaging the urban landscape, this study was conducted to accurately identify the species and trace their route of introduction into Korea. Based on distinct morphological features and gene-based intra- and interspecific divergence analyses, the pest group was identified as two species, *Sphecodoptera sheni* and *Paranthrenella pinoakula* sp. n. To track their movement in Korea, haplotype analyses revealed that *S. sheni* is likely to have spread to five other neighboring areas, mainly in the Wonju and Goyang populations, while *P. pinoakula* sp. n. could only be identified in Wonju. Furthermore, the phylogenetic tree confirmed that *P. pinoakula* sp. n. is closely divergent from congeneric species and distantly related to *S. spheni*. Accurate species identification of pests is paramount to pest management strategies, and tracking migration routes is essential research for ongoing surveillance and pre-emptive action in high-infestation areas.

**Abstract:**

The populations of clearwing moth borers in Korea have recently caused extensive and severe damage to pin oaks (*Quercus palustris* Munchh.). We conducted field monitoring and molecular analyses to identify them in an integrated manner. Morphological examination and molecular analyses of the COI gene, based on intra- and interspecific genetic divergences (GDs), revealed that the borers were identified as two invasive species, *Sphecodoptera sheni* and *Paranthrenella pinoakula* sp. nov. The maximum intraspecific GD was found to be 1.9%, whereas the minimum interspecific GD was confirmed as 8.1%, indicating a distinct barcoding gap. Both the MJ network and NJ tree also showed that 18 haplotypes (Hs) were detected from the 52 COI sequences. The borers revealed a total of 17 Hs: (i) H1–H7 were detected in all seven regions with *S. sheni*; (ii) Wonju and Goyang populations of *S. sheni* revealed more than three Hs; (iii) H7 was closely connected with H8 of the Chinese population of *S. sheni*; (iv) H9–H10 were detected in other samples from the Wonju population with *P. pinoakula* sp. n. and were closely located with congeneric species. A maximum likelihood tree also revealed that *P. pinoacula* sp. n. nested within the congeneric species, genetically separating from *S. sheni*.

## 1. Introduction

The family Sesiidae (Lepidoptera: Cossoidea), established by Boisduval in 1828 [1], is distributed worldwide, with a majority found in the Afrotropical and Oriental realms [2]. To date, more than 1563 species of 171 genera have been recorded [3]. Sesiidae comprises medium- to large-sized moths that have an unusual appearance. Their transparent wings, unlike other moths, are not covered with scales, earning them the name ‘Clearwing moths’. Additionally, adults exhibit superficial characteristics resembling hymenopterans, such as wasps and hornets, making them a prime example of Batesian mimicry [4]. The larvae of the sesiids are typically obligate borers, creating holes in tree trunks, branches, fruit, or roots. They feed on the phloem and cambium by burrowing inward. Their feeding behavior causes significant damage to commercially and economically important agricultural and forestry fruit trees, such as *Malus pumila* Mill., *Prunus persica* (L.) Batsch, *Vitis vinifera* L., *Diospyrous kaki* Thunb., and *Zizyphus jujuba* Mill. [5,6,7,8,9,10].

Starting with Matsumura’s reports, which included 4 species, *Entrichella constricta* (Butler), *Scalarignathia coreacola* (Matsumura), *Synanthedon quercus* (Matsumura), and *S. tenuis* (Butler) [11,12], a total 24 species belonging to 13 genera (*Bembecia*, *Glossosphecia*, *Macroscelesia*, *Melittia*, *Nokona*, *Scasiba*, *Scalarignathia*, *Sesia*, *Synanthedon*, *Entrichella*, *Milisipepsis*, *Paranthrenopsis,* and *Pennisetia*) of the Sesiidae have been recorded in Korea [13]. This number is relatively small compared to the 45 species recorded in Japan [14,15] and the 108 species recorded in China [16]. Research on Korean Sesiidae has mainly focused on fauna checklist [5], taxonomic description [6,7,8], and pheromone control [9,10] of pests on fruit trees.

There has been a request for monitoring surveys on clearwing moths in Korea for several years now. The expansion of domestic fruit tree cultivation, driven by global warming, has led to the clearwing moth pest expanding its distribution area and causing extensive damage to adjacent forestry or street trees in urban areas. One example is *Synanthedon bicignulata*, a pest of peach, plum, apricot, and other fruit trees, which has recently expanded its distribution beyond agricultural areas to urban centers as a pest that causes severe damage to yoshino cherry (*Prunus yedoensis* Matsumura), an urban street tree [17].

Recently, clearwing moth borer damage has been frequently reported in pin oak street trees in Korea. The pin oak, *Quercus palustris* Munchh. (Fagaceae), is one of the most popular varieties to be planted as a domesticated street tree in Korea since the early 2000s (Figure 1). The emergence of unknown pests on urban street trees has caused citizens to feel uncomfortable and anxious that they may carry diseases. Online sources have listed some clearwing moths, such as *Scasiba rhynchioides* (Butler), *Synanthedon bicingulata* (Staudinger), and *S. quercus* (Matsumura), as pin oak pests, but there has been no professional examination and review at the species level.

For the last two years, our research team has been monitoring clearwing moth borers that attack pin oaks planted as street trees nationwide. We observed that multiple pin oak populations have significant pest damage signs throughout Korea. The congruent symptoms were caused by the larvae, which bore holes in the xylem, crawl inside the tunnels to feed, and ooze milky sap with coarse reddish-brown excrement on the bark of pin oaks. The larvae attack the cambium layer of the xylem structure and create a pupal frame under barks. This leads to stunted tree growth and can cause decline. In addition, emerged adult moths leave conspicuous marks, including pupal exuviae and holes on the bark that disturb the urban landscape (Figure 2A–F). These larvae are often confused with beetle larvae due to the similarities in perforating feeding habits, large body size, and milky coloration. However, the number of prolegs, development of pinaculums, and other characteristics allow them to be classified as clearwing moth larvae (Figure 2G).

The study has two objectives. Firstly, we monitor and identify the clearwing moth borer attacking pin oak street trees in Korea based on morphological and molecular analyses. Both external characteristics and internal morphology, including genitalia, will be examined, and for molecular identification, genetic divergences within and between species will be analyzed using COI genes, which are useful for lepidopteran species delimitation [18].

Secondly, we intend to predict the domestic routes of migration of the clearwing moth borers within Korea. To achieve the research objectives, we analyzed the median-joining network about haplotypes and constructed a neighbor-joining (NJ) tree referring to genetic distance. Furthermore, phylogenetic analysis was used to infer the relationships between the clearwing moth pest groups.

## 2. Materials and Methods

### 2.1. Sampling from the Field

From 2021 to 2023, 41 individuals were collected from pin oak street trees in seven regions (Bucheon, Goyang, and Seongnam of Province Gyeonggi (GG); Chuncheon and Wonju of Prov. Gangwon (GW); Cheongju of Prov. Chungcheongbuk (CB); Daegu Metropolitan City and Gwangju Metropolitan City) of Korea. Among the regions, the pin oaks in Goyang and Wonju were planted in the early 2000s, while the others were planted within 5 years. In Gwangju Metropolitan City, we have only confirmed the clearwing moth borer damage, but no individuals were caught.

Since they were in larval or pupal stages, they were reared in a falcon tube (200 mL) individually to adults and maintained at 26–27 °C, 12:12 (L:D), 60% humidity in the laboratory.

For morphological identification, all individuals were examined for their external characteristics and photographed under a Leica Z16 APO stereomicroscope and Dhyana 400DC (4M) sCMOS camera (TUCSEN, Fuzhou, China) with Mosaic Analysis Software 2.4. Genital slide vouchers were made according to Kim et al. [19]. All specimens and slide vouchers were deposited in the laboratory of Insect Phylogenetics and Evolution, Jeonbuk National University (IPE JBNU), Korea.

### 2.2. Molecular Protocol

A total of 52 sequences were included for molecular protocol. Of them, 50 sequences were used as ingroup: (i) 9 sequences of 6 species, *Paranthrenella auriplena* (Walker, [1865]), *P. cinnamoma* Yu, Gao, Kallies, Arita and Wang, 2021, *P. chrysophanes* (Meyrick, 1887), *P. formosicola* Strand, 1916, *P. terminalia* Kallies, 2020, and *Sphecodoptera sheni* (Arita and Xu, 1994), from BOLD public data (BOLD:ABW9347, BOLD:AAI7641, BOLD:ACO5543, BOLD:AEN4976, BOLD:AEY6232, BOLD:ACO5542); (ii) 41 sequences of 2 species (*Sphecodoptera sheni* and *Paranthrenella pinoakula* sp. n.) from field survey were newly added (GenBank accession no. PP003748-PP003788; this study). Two sequences of one species (*Anatrachyntis japonica* Kuroko: Cosmopterigidae) used as the outgroup were also newly added (GenBank accession no. PP003789-PP003790; this study). Information (sample ID, sequence database no., and distribution) for all sequences used in molecular analyses is provided in Appendix A.

DNA was extracted from the head or leg of each specimen using a Tissue Genomic DNA Isolation Kit Mini in LaboPass™ DNA Purification Kit (Cosmo Genetech Co. Ltd., Seoul, Republic of Korea), following the manufacturer’s protocol. For PCR amplification for mitochondrial cytochrome oxidase subunit I (653 bp), the universal primer set, LCO 1490 (5′-GGTCAACAAATCATCATAAAGATATTGG-3′) and HCO 2198 (5′ GCTACAACATAATAAGTATCATG-3′) [20], was used under the following conditions: initial denaturation at 94 °C for 5 min, followed by 34 cycles of denaturation at 95 °C for 1 min, annealing at 45.2 °C for 1 min, extension at 72 °C for 1 min, and a final extension at 72 °C for 5 min. Amplification was performed using AccuPower PCR Premix (Bioneer, Daejeon, Republic of Korea) following the standard protocols. The PCR products were checked in 1.2% agarose gels, purified, and sequenced at Macrogen, Inc. (Geumcheon-Gu, Seoul, Republic of Korea).

### 2.3. Data Analysis

Intra- and interspecific pairwise genetic distance and neighbor-joining (NJ) tree were conducted using MEGA 7.0 under the Kimura 2-parameter model [21]. To determine the evolutionary relationships among haplotypes of the sesiid species, the median-joining (MJ) network was analyzed in Network ver. 10.2.0.0. A phylogenetic tree was constructed based on the haplotypes of Sesiinae detected in this study. The maximum likelihood (ML) tree was implemented by IQ-Tree 2.1.1 [22] with the best-fit model, GTR+F+G4, according to the Bayesian information criterion [23,24]. Branch support values were produced by ultrafast bootstrap pseudoreplicates (UFB) [25] and a Shimodaira–Hasegawa approximate likelihood ratio test (SH-aLRT) [26]. Every test was with 100,000 pseudoreplicates each for UFB and SH-aLRT.

## 3. Results

### 3.1. Morphological Identification

Morphologically, two species of sesiid borers that damage pin oaks were recognized based on adult and genitalic characteristics. These species were respectively placed in the genera Sphecodoptera and Paranthrenella, which were previously unrecorded in Korea. The two sesiid borers were identified as *Sphecodoptera sheni* and *Paranthrenella pinoakula* sp. nov.

#### Taxonomic Accounts

**Genus *Sphecodoptera* Hampson, [1893]** [27].*Sphecodoptera;* Hampson, [1893] [27]: 189. Type species. *Sphecia repanda* Walker.*Sphecia*; Hampson, 1919 [28]: 80.*Sesia*; Spatenka, Lastuvka, Gorbunov, Tosevski, and Arita, 1993 [29]: 87.*Spherodoptera;* Matsumura, 1931 [11]: 1017.*Scasiba;* Matsumura, 1931 [30]: 8.

***Sphecodoptera sheni* (Arita and Xu, 1994)** [30]*Sesia sheni;* Arita and Xu, 1994 [30]: 61. Type locality. Nanjing, Jiangsu, China.*Scasiba caryavora*; Xu and Arita, 1994 [31]: 2.*Sphecodoptera sheni*; Kallies, Arita, Owada. and Wang, 2014 [32]: 587.

Material examined. Korea. GW: 2♀, Wonju, Moosil-dong 1710, 8.VII.2020, J-K. Jung; 2♂, 1♀, ditto, 12.VII.2022, J-K. Jung, gen. slide no. IPE04, IPE05/S. Kim; 5 exs, Chuncheon, Woodoo-dong 1121, 12.V.2023, B. Lee. GG: 9 exs, Seongnam, Bungdang-gu, Sampyong-dong, 15.VII.2022, J-K. Jung; 5, Goyang, Ilsan, Juyeob-dong 152, 6.V.2023, B. Lee; 6, Bucheon, Yeokgok-dong, 3.VI.2023, B. Lee. CB, 2, Chungju, Heungdeok-gu, 28.IV.2023, S. Kim. Daegu, 6, Jung-gu, Gukchaebosang-ro 139gil 1, 8-12.IV.2023. I. Park.

Diagnosis. This species is superficially similar to *Scasiba rhynchioides* in having a large-sized body and adult coloration (Figure 3), but it can be easily recognized by the genitalic characteristics. The male genitalia of *S. sheni* are distinguished from the latter by the finger-shaped processus of the saccular margin antemedially (Figure 4E). The processus of the saccular margin of *S. rhynchioides* is like a triangular horn, wide at the base and narrowing toward the apex (see Figures 3 and 4 in Arita and Fukuzumi, 1988 [33]).

The female genitalia of *S. sheni* are also differentiated from the latter by the long and narrow ovate corpus bursae bearing a distinct signum, large and spine shaped (Figure 4G,H). The female genitalia of *S. rhynchioides* have a small and circular-shaped signum (see Figure 70 in the master dissertation of Lee, 2003 [34]).

Host Plants. *Carya illinoensis* (Wang.) K. Koch (Juglandaceae); *Quercus suber* Linnaeus, *Q. variabilis* Blume, *Q. acutissima* Carruthers, and *Castanea mollissima* Blume (Fagaceae) [30,35]; *Quercus palustris* Münchh. (this study).

Distribution. Korea (GG, GW, CB, Daegu) (this study), China (Jiangsu) [30].

Remark. Of the pin oak street borers, the large-sized sesiids were identified as *S. sheni* in this study. This species was found in Korea for the first time since it was reported in China. According to Arita et al. [30], this species has been reported as a pest of North American native pecan trees planted as street trees in Nanjing, China. Based on the investigation of the number of pupal exuviae, *S. sheni* was higher than other sessid borers in density, and their distribution on trees was aggregated between ground to 1 m in height.

**Genus *Paranthrenella* Strand, 1916** [36].*Paranthrenella* Strand, 1916 [36]: 47. Type Species. *Paranthrenella formosicola* Strand.


***Paranthrenella pinoakula* sp. nov. Kim**


Type Material. Holotype: One male, Korea, GW, Wonju, Moosil-dong 1710, 2.VII.2022, J-K. Jung, gen. slide no. IPE03/S. Kim; paratypes: two males, ditto, 8.VII.2020, J-K. Jung.

Diagnosis. This species is externally similar to *Paranthrenella formosicola* in having a long and slender abdomen (Figure 4E). However, it can be readily distinguished from the former due to the lesser development of yellow scales between the thorax and tegula (Figure 4C), as well as the abdomen dorsally (Figure 4E). Except for the fourth and seventh tergite, the presence of yellow scales on the posterior margin of each tergite is not noticeable. In contrast, *P. formosicola* has thickly developed yellow markings on the posterior margin of the second, fourth, and sixth tergites in males and almost entire tergites in females. The male genitalia are also differentiated from that by the asymmetrical valva, including a ridge line and aedeagus bearing four small cornuti at the apex.

Description. Male (Figure 5). Wingspan: 22.0–28.0 mm; forewing length: 10.0–12.5 mm; body length: 17.0–19.5 mm. Head: frons pale brown; vertex dark brown; occipital fringe yellow; scape of antenna yellow entirely, shorter than 1/2 diameter of eye; flagellum blackish brown antemedial and apical parts, tinged with yellow postmedial part dorsally, short setose at apex, yellow ventrally; labial palpus entirely yellow; third segment of labial palpus 2/3 length of second segment. Thorax: patagium blackish brown, mixed with pale brown laterally; tegula blackish brown with yellow scales on posterior margin; mesothorax blackish brown with two spots of yellow scales on posterior margin; metathorax blackish brown. Forewing hyaline; costal margin blackish brown; dorsal margin yellowish brown; wing base blackish brown tinged with bright yellow; bright yellow between radius vein 2 (R2) and R3; cilia yellowish brown; forewing vernation: radius 4 (R4) stalked with R5 antemedially; anterior transparent area (ATA) extended to 3/4 of forewing; posterior transparent area (PTA) extended to 4/5 of forewing; apical 1/3 exterior transparent area tinged with yellowish brown scales. Abdomen blackish brown with yellow scales noticeable at the posterior margin of tergite 4 (T4) and T7.

Male genitalia (Figure 6). Tegumen–uncus complex broad; scopula androconialis well developed, slightly shorter than the length of tegumen–uncus complex; crista gnathi lateralis large, cup shaped; crista gnathi medialis relatively narrow, oval. Valvae elongated, oval, asymmetrical; right valva with asymmetric ridge line from basal crista sacculi to apex, rather dense setose followed the ridge from base to 1/3 of valva; left valva without ridge. Saccus with bifurcate terminal margin, reverse blunt fork shaped, as long as vinculum. Aedeagus as long as valva, gradually narrow from base to apex; vesica curved at sub-apex; bearing four cornuti.

Female genitalia. Unknown.

Host Plants. *Quercus palustris* Münchh. (this study).

Distribution. Korea (GW) (this study).

Etymology. The species name is derived from a common name of the host plant (*Quercus palustris*), pinoak- plus a Latin diminutive suffix, -ula, referring to the insect pest borer of the pin oak.

*Remark.* In the pin oak street insect pests, the small-sized borers were identified as a new species, *P. pinoakula* sp. n., in this study. This species was only found in Wonju (GW) in Korea, and their densities were relatively lower compared to *S. sheni*.

The genus *Paranthrenella*, which includes the new species, is mainly distributed in Oriental and Australasian regions, comprising 20 species [2,37]. The discovery of this species in Korea, which is not zoogeographically part of the Oriental region, suggests a high probability of its establishment as an invasive species.

### 3.2. Molecular Analyses

#### Genetic Divergence and Distribution

Genetic divergences within all species included in this study are presented in Table 1. Except for four species (*Paranthrenella auriplena*, *P. cinnamoma*, *P. formosicola*, and *P. terminalis*), which have only one sequence, a total of 647 comparison pairs of four species (*S. sheni*, *P. pinoakula* sp. n., *P. chrysophanes*, and *Anatrachyntis japonica*) were calculated within the species. The intraspecific genetic distance (GD) averaged 0.3%, and the maximum intraspecific divergence did not exceed 1.9%. These of *S. sheni* and *P. pinoakula* sp. n. were 1.7% and 0.3%, respectively. The highest value of intraspecific GD was found in *P. chrysophanes*, with 1.9%.

The interspecies GD was analyzed with 564 comparison pairs. The minimum interspecific GD among all seven species was calculated to be 8.1% between *P. auriplena* and *P. chrysophanes*, while the maximum interspecific genetic value was 16.8% (averaging 13.8%) between *P. terminalis* and *P. chrysophanes*. The distribution of their genetic divergence is shown in Figure 7. All species analyzed in this study exhibited a distinct barcode gap between the maximum intraspecific GD (1.9%) and the minimum interspecific GD (8.1%) in the COI gene.

### 3.3. NJ Tree Analysis

Neighbor-joining (NJ) tree analysis, a heuristic method based on genetic distance, showed that *S. sheni* and *P. pinoakula* sp. n. branched into clades A and B, respectively (Figure 8). In clade A, the Chinese *S. sheni* individual was the first to branch out, followed by the Wonju (GW) population. And then, each population of Goyang (GG) and Chuncheon (GW) branched off, followed by the populations of Bucheon (GG) and Cheongju (CB), and all diverse populations were claded together. All individuals of *P. pinoakula* sp. n. were nested within the congeneric species (*P. terminalia*, *P. formosicola*, *P. cinnamoma*, *P. auriplena*, and *P. chrysophanes*) in clade B. Of them, the *P. pinoakula* sp. n. was also genetically close to *P. formosicola,* which was the allied species with similar morphology.

### 3.4. Haplotype Analysis

The median-joining network (MJ) analysis presented that 18 haplotypes were detected from the 52 COI sequences (Figure 9). Haplotype distribution information for all individuals is provided in Appendix A.

A total of 36 sequences of Korean clearwing borers revealed a total 10 haplotypes. H1–H8 haplotypes of 34 sequences were recognized as *Sphecodoptera sheni* and H9–H10 haplotypes of three sequences were revealed as *Paranthrenella pinoakula* sp. n.

H1–H7 haplotypes were detected from Korean samples (BC, GY, SN, CC, WJ, CJ, DG) with large-sized borers, *S. sheni*. WJ and GY populations of *S. sheni* were revealed as three haplotypes, and among them, the H7 haplotype (WJ) was closely connected with the H8 haplotype of Chinese *S. sheni*. BC and CJ populations of that were presented as two haplotypes, H1 and H2 haplotypes. The CC population was recognized as two haplotypes, H1 and H3 haplotypes, whereas SN and Daegu populations were detected in only the H1 haplotype. H9 and H10 haplotypes were detected as the other Korean Wonju population of small-sized borers, *P. pinoakula* sp. n., and closely located with each congeneric species (*P. terminalia*, *P. formosicola*, *P. cinnamoma*, *P. auriplena,* and *P. chrysophanes*).

### 3.5. Phylogenetic Tree

Although one gene is not enough to analyze a phylogenetic tree, a maximum likelihood tree (MLtree) was constructed to show the phylogenetic relationship between the species and individual at the species level. The MLtree is derived from a statistical method based on evolutionary models, including assumptions about the probability distribution of the data. As a result, all individuals of the large-sized clearwing moth borer, *Sphecodoptera sheni,* were separated from the other *Paranthrenella* spp. (Figure 10, clade A: Sh-aLRT/UFB/PP: 98.8/100/100). The small-sized clearwing moth borer, *P. pinoakula* sp. nov., was also phylogenetically claded within the genus *Paranthrenella* (Figure 10, clade B: Sh-aLRT/UFB/PP: 100/100/100). The species is closely located in the clade of *P. formosicola* and *P. terminalia.* Two borers are phylogenetically distantly related.

## 4. Discussion

Climate change presents a global threat to humanity [38,39]. Certain human activities, such as the expansion of worldwide trade and the increase in domestic and international travel, have led to unanticipated issues, such as invasive species. Since 1900, 89 different invasive species have appeared in Korea, with 33 of them, or 38% of the total, introduced in the last 19 years [40]. It appears that the introduction of exotic pests into the country is accelerating as a result of global warming. Moreover, extreme weather phenomena are not only spreading exotic pests but also triggering outbreaks of native insect pests in human habitats and forests.

While the ecological function of urban street trees is to provide natural scents and shade that act as a carbon sink in urban areas, they also serve as a special habitat for invasive forest pests. This study is a case of unexpected insect pest outbreaks of street trees planted for the purpose of urban landscaping and environmental clean-up. Many cases have been reported in the past few years where planted pin oak street trees were covered with milky sap and excrement, and the appearance of larvae caused disgust among citizens. Our research team conducted this study to determine the cause of this phenomenon.

The purpose of this study is two-fold. The first objective is to identify the recent outbreak of clearwing moth borers attacking pin oaks, one of the popular street tree species planted in Korea. The pin oak borers are actually referred to as a variety of clearwing moths, e.g., *Scasiba rhynchioides* (Butler), *Synanthedon bicingulata* (Staudinger), and *S. quercus* (Matsumura), which are similar in appearance. Using morphological identification-based genitalic characteristics, pin oak street tree borers were identified as *Sphecodoptera sheni* and *Paranthrenella pinoakula* sp. nov. We unexpectedly confirmed that there are two invasive species and that they are distinguished by body size. COI gene-based molecular identification also supported the two clearwing borers, showing a distinct barcode gap within and between species.

*Sphecodoptera* and *Paranthrenella,* which include both pests affecting pin oak street trees in Korea, have never been recorded in the country. Species diversity within these genera is high in the Oriental regions, especially Southeast Asia. Korea is zoogeographically located in the Palearctic region, making it highly likely that these pests are exotic. Most invasive pests introduced into Korea have been found in coastal areas, particularly in the southwestern (Jeollabuk-do and Jeollanam-do) and southern (Is. Jeju) parts of Korea. They are often carried by prevailing westerly winds originating from Southeast Asia through the eastern coast of China, or some have been found around ports or in border quarantine areas. In the case of *S. sheni*, this is the only record from Korea since its announcement as a new species in eastern China (Jiangsu) in 1994. The new species, *P. pinoakula* sp. n., is also likely to be an invasive pest, given the high species diversity of the genus in Southeast Asia.

To predict the domestic migration routes of the two pests, we derived an NJ tree based on genetic distance between individuals and conducted haplotype analysis based on MJ network analysis to identify genetic variation among local populations.

The results of the NJ tree and haplotype analysis suggest that *S. sheni* may expand its distribution from China. The outbreak was centered in WJ (GW) and GY (GG), each with three haplotypes, to neighboring areas of CC (GW), BC (GG), and CJ (CB), each with two haplotypes. Relatively, both SN (GG) and Daegu populations are detected to have the H1 haplotype, inferring a recent outbreak. On the other hand, for *P. pinoakula* sp. nov., only the WJ (GW) population was confirmed. Furthermore, MLtree confirmed that the two borers are phylogenetically distantly related.

Consequently, the two clearwing moth borers that damaged the pin oak street trees were identified as *S. sheni* and *P. pinoakula* sp. n., both of which are likely to be invasive pests and are not closely related phylogenetically, but rather groups that share the same habitat due to feeding preferences. In addition, it can be seen as a case where street trees in urban areas serve as a habitat for invasive pests.

## Figures and Tables

**Figure 1 insects-15-00079-f001:**
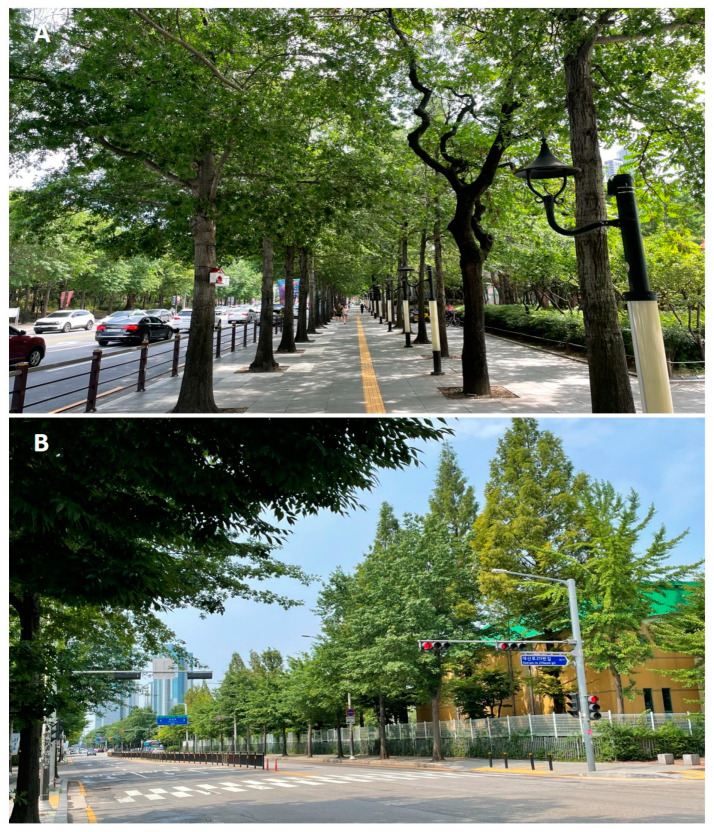
Panoramic view of pin oak (*Quercus palustris* Munchh.) street trees. (**A**) Daegu metropolitan city; (**B**) Ilsan-gu, Goyang-si, Province Gyeonngi.

**Figure 2 insects-15-00079-f002:**
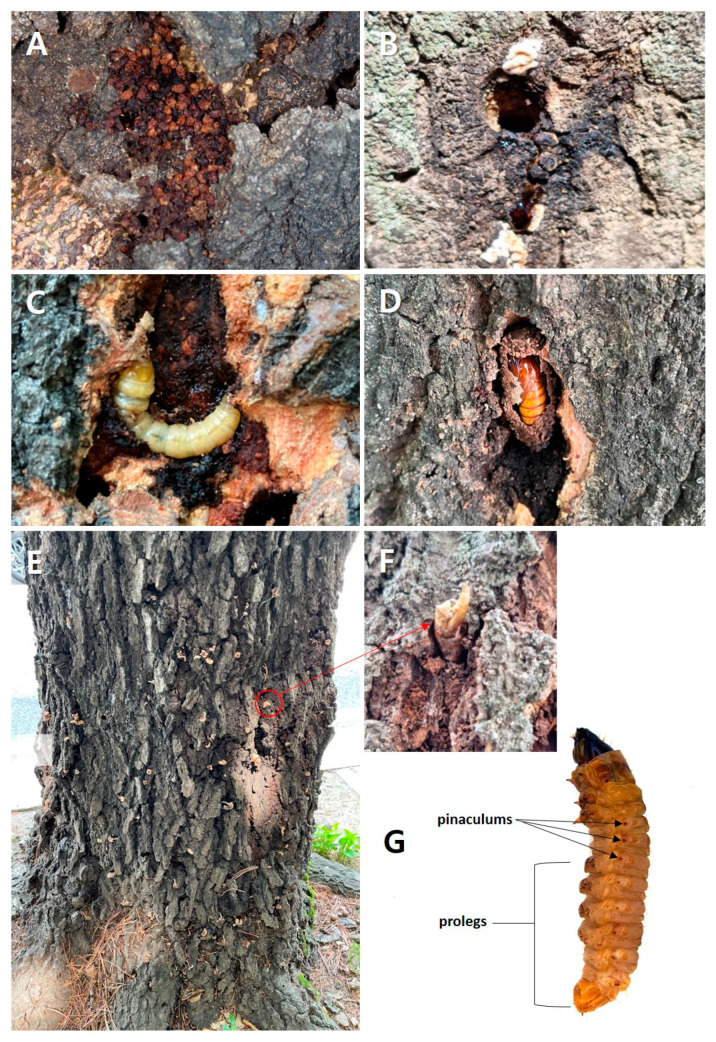
Diverse damage on pin oak street trees by clearwing moths. (**A**) Reddish-brown frass by larva; (**B**) milky sap damage from cavity in xylem made by larva; (**C**) xylem cambium eaten by larva; (**D**) pupal frame in cambium; (**E**) distinctive marks left on the bark after emergence; (**F**) zoom in on exuviae after larva emergence; (**G**) clearwing moth larva.

**Figure 3 insects-15-00079-f003:**
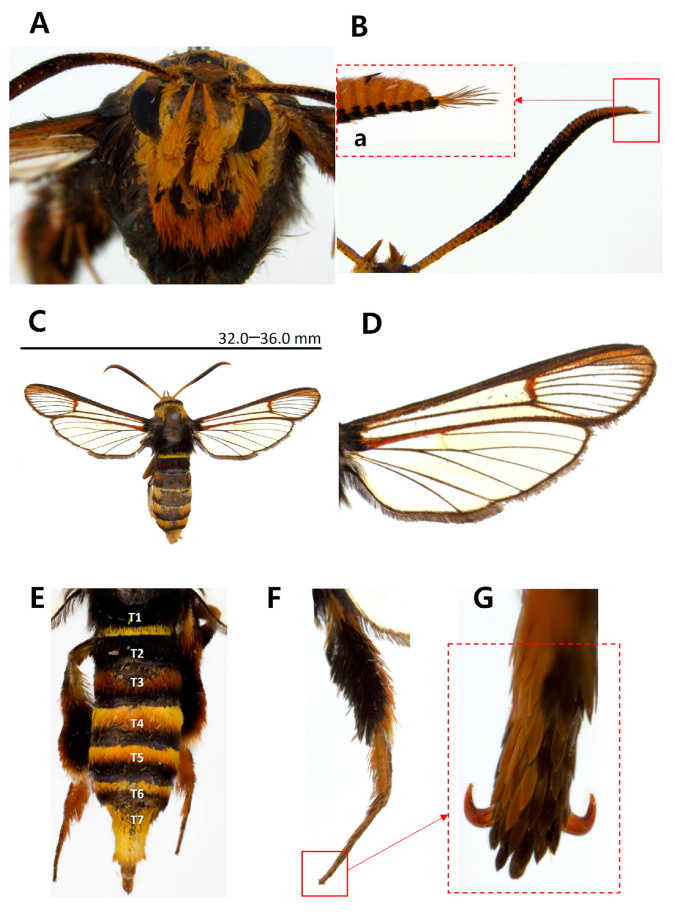
Adult of *Sphecodoptera sheni*. (**A**) Head; (**B**) flagellum of antenna; (**Ba**) apical part of flagellum; (**C**) entire view; (**D**) wings; (**E**) abdomen; **T1–T7**, 1st tergite to 7th tergite; (**F**) hind leg; (**G**) tarsal claw of hind tarsus; scale bar indicates wingspan.

**Figure 4 insects-15-00079-f004:**
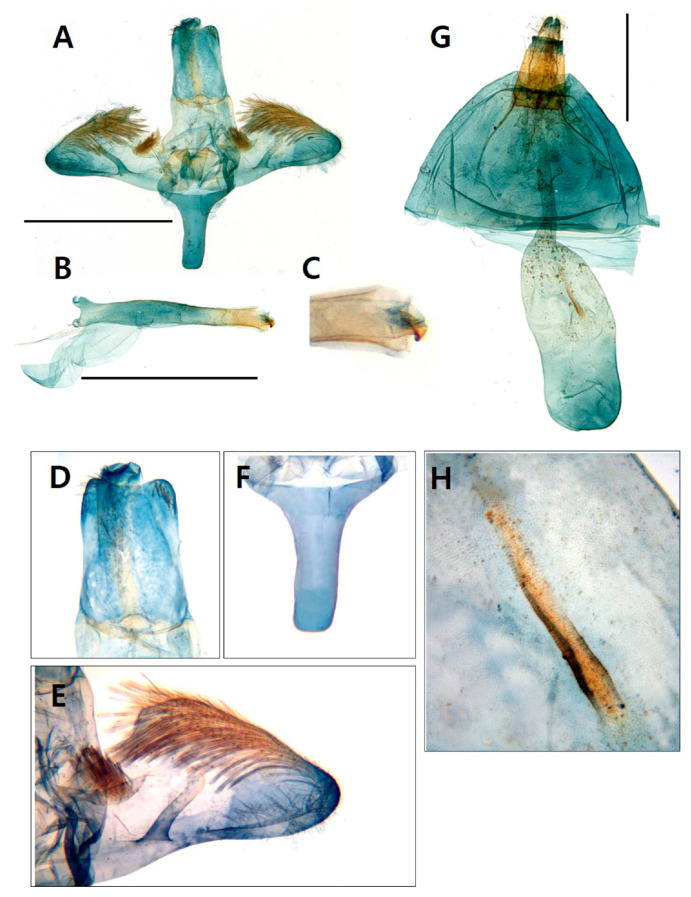
Genitalia of *Sphecodoptera sheni.* (**A**) Male genitalia; (**B**) aedeagus; (**C**) apical part of aedeagus including cornutus; (**D**) uncus; (**E**) a finger-like processus on the saccular margin of valva; (**F**) saccus; (**G**) female genitalia; (**H**) signum in corpus bursae; scale bar: 0.25 mm.

**Figure 5 insects-15-00079-f005:**
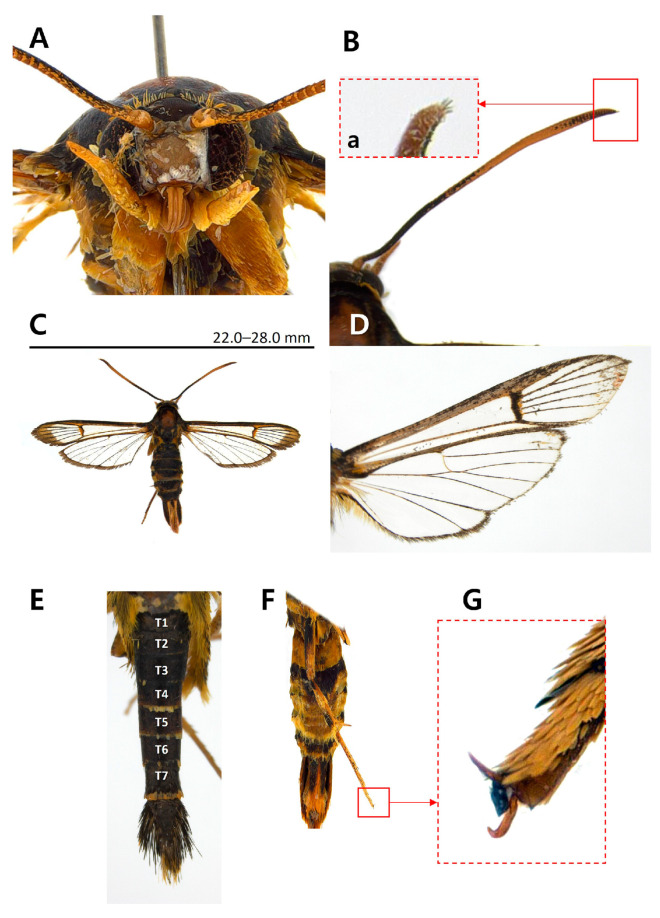
Adult of *Paranthrenella pinoakula* sp. nov. (**A**) Head; (**B**) flagellum of antenna; (**Ba**) apical part of flagellum; (**C**) entire view; (**D**) wings; (**E**) abdomen; **T1–T7**, 1st tergite to 7th tergite; (**F**) hind leg; (**G**) tarsal claw of hind tarsus; scale bar indicates wingspan.

**Figure 6 insects-15-00079-f006:**
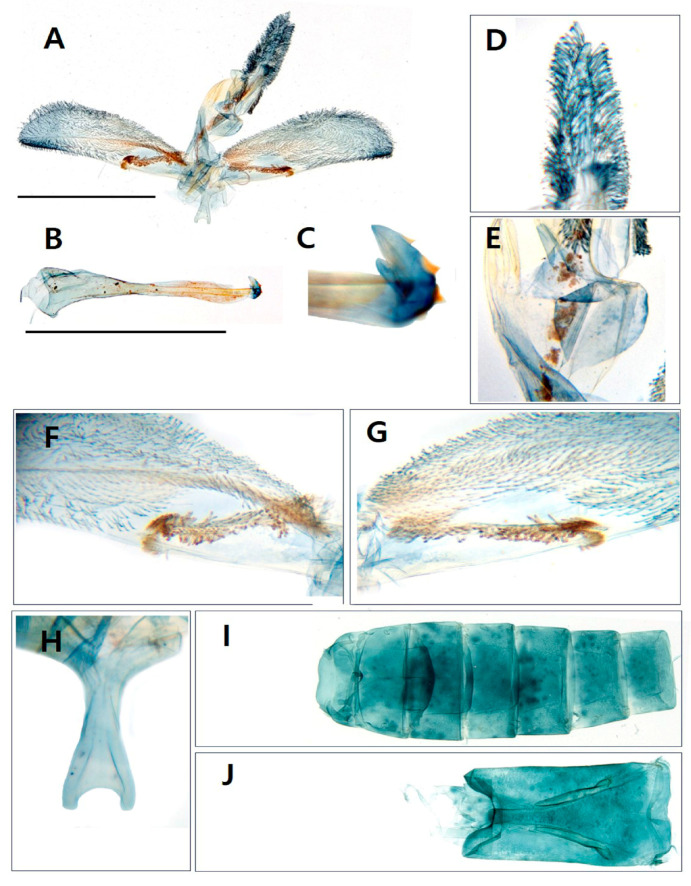
Genitalia of *Paranthrenella pinoakula* sp. nov. (**A**) Male genitalia; (**B**) aedeagus; (**C**) apical part of aedeagus including cornute; (**D**) scopula androconialis; (**E**) medial gnathos flap; (**F**) right valva with a ridge line; (**G**) right valva; (**H**) saccus; (**I**) sternum I-VII segments; (**J**) sternum VIII segment; scale bar: 0.25 mm.

**Figure 7 insects-15-00079-f007:**
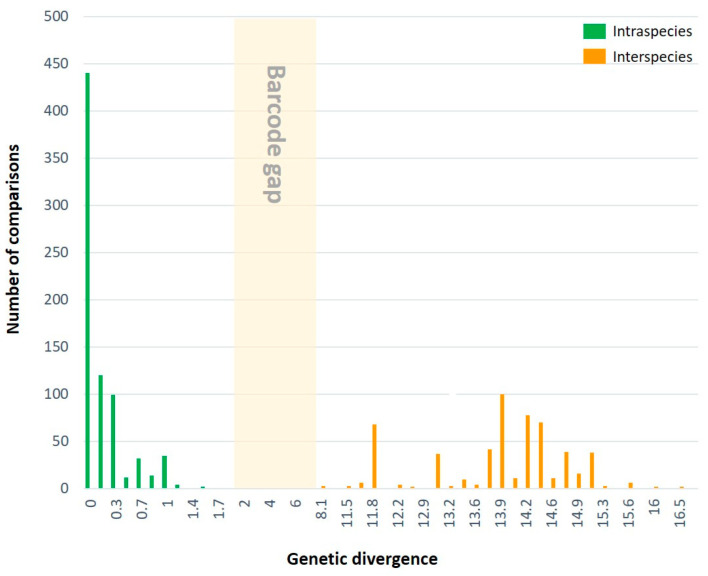
Distribution of genetic divergences based on the Kimura-2 parameter model for COI sequences according to taxonomic levels. A total of 647 comparison pairs within the species and 564 comparison pairs between the species were calculated.

**Figure 8 insects-15-00079-f008:**
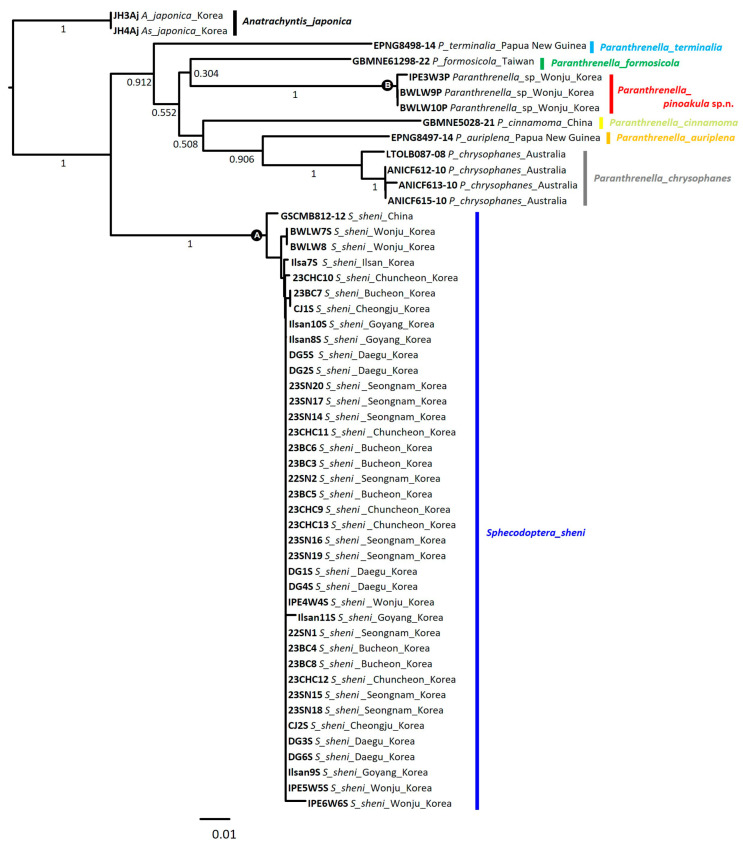
Neighbor-joining tree of COI gene of Sesiid species. Numbers under the branches are bootstrap percentages (%). The scientific name and distribution information for each individual are appended to the sample ID.

**Figure 9 insects-15-00079-f009:**
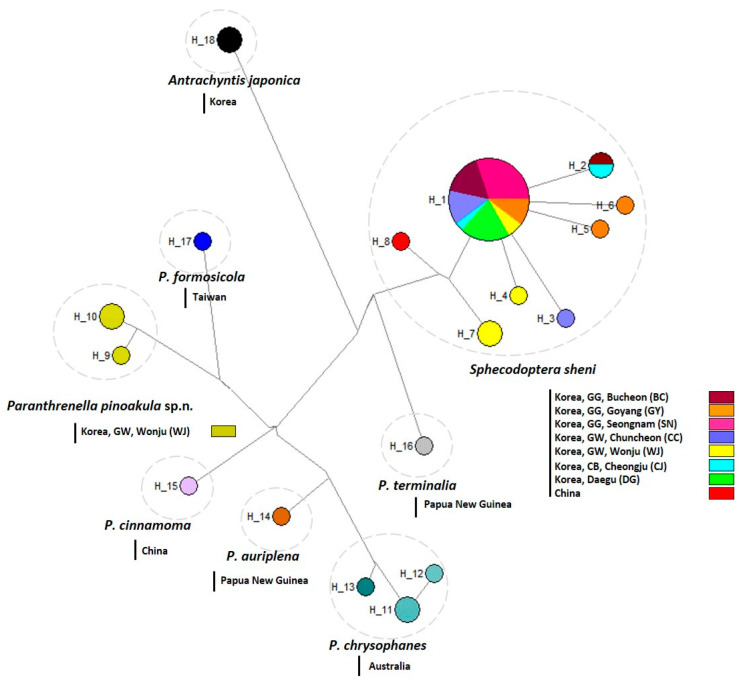
Median-joining network of 18 haplotypes of the 52 COI sequences. The pie size is proportional to the haplotype frequency. A dotted circle means the same species.

**Figure 10 insects-15-00079-f010:**
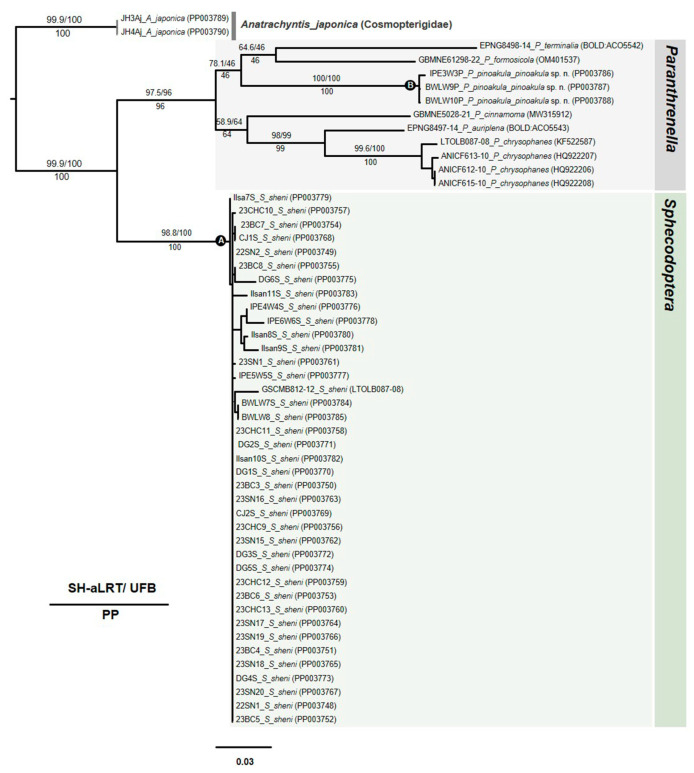
Maximum likelihood tree of Sesiid species. Each node marked with three types of supporting value: Shimodaira–Hasegawa approximate likelihood ratio test (SH-aLRT): upper left; ultrafast bootstrap (UFB) pseudoreplicates: upper right; maximum likelihood (ML) bootstrap: below the branch. Each scientific name is followed by the sample ID, and Genbank accession number (or BOLD number) is indicated in parentheses.

**Table 1 insects-15-00079-t001:** Genetic divergences within all species included in this study.

**(1) Different taxonomic levels**				
**Taxonomic level (No. of Comparison Pairs)**		**K2P Pairwase distances**		
		Maximum	Minimum	Average
Within species (647)		1.9	0	0.3
Between species (564)		16.8	8.1	13.8
**(2) Intraspecific genetic divergence**				
**Species**	**Comparison pairs** **(CP)**	**Intraspecific genetic divergence**		
		Maximum	Minimum	Average
** *S. sheni* **	637	1.7	0	0.2
***P. pinoakula* sp.n.**	3	0.3	0	0.2
** *P. auriplena* **	-	-	-	-
** *P. cinnamoma* **	-	-	-	-
** *P. chrysophanes* **	6	1.9	0	0.9
** *P. formosicola* **	-	-	-	-
** *P. terminalis* **	-	-	-	-
** *A. japonica* **	1	0	0	0
**(3) Interspecific genetic divergence**				
**species**	** *S. sheni* **	***P. pinoakula* sp. n.**	** *P. auriplena* **	** *P. cinnamoma* **
** *S. sheni* **		CP = 114, 14.6 (14.2–15.3)	CP = 38, 14.2 (13.9–14.9)	CP = 38, 15.1 (14.9–15.8)
***P. pinoakula* sp.n.**			CP = 3, 13.9 (13.9–13.9)	CP = 3, 13.0 (13.0–13.0)
** *P. auriplena* **				CP = 1, 11.5 (11.5–11.5)
** *P. cinnamoma* **				
** *P. chrysophanes* **				
** *P. formosicola* **				
** *P. terminalis* **				
** *A. japonica* **	CP = 78, 11.8 (11.5–12.5)	CP = 6, 14.9 (14.9–14.9)	CP = 2, 13.7 (13.7–13.7)	CP = 2, 15.6 (15.6–15.6)
**species**	** *P. chrysophanes* **	** *P. formosicola* **	** *P. terminalis* **	
** *S. sheni* **	CP = 152, 14.0 (13.7–14.9)	CP = 38, 13.1 (12.9–13.7)	CP = 38, 13.7 (13.4–14.4)	
***P. pinoakula* sp.n.**	CP = 12, 13.4 (13.0–13.7)	CP = 3, 11.8 (11.7–12.0)	CP = 3, 14.3 (14.2–14.4)	
** *P. auriplena* **	CP = 4, 8.2 (8.1–8.4)	CP = 1, 13.2 (13.2–13.2)	CP = 1, 14.9 (14.9–14.9)	
** *P. cinnamoma* **	CP = 4, 138 (13.7–13.9)	CP = 1 12.2 (12.2–12.2)	CP = 1, 14.4 (14.4–14.4)	
** *P. chrysophanes* **		CP = 4, 14.4 (14.2–14.6)	CP = 4, 16.5 (16.3–16.8)	
** *P. formosicola* **			CP = 1, 12.2 (12.2–12.2)	
** *P. terminalis* **				
** *A. japonica* **	CP = 8, 15.6 (15.3–16.0)	CP = 2, 14.9 (14.9–14.9)	CP = 2, 14.1 (14.1–14.1)	

## Data Availability

The following information was supplied regarding the availability of molecular data: COI sequences are deposited in GenBank of NCBI, and the accession numbers are PP003748-PP003788.

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
