# Peer review of "Integrated Identification and Genetic Diversity of Potentially Invasive Clearwing Moths (Lepidoptera: Cossoidea: Sesiidae) in Korea"

_insects, 2024, doi:10.3390/insects15010079_

Round 1

Reviewer 1 Report

Comments and Suggestions for Authors

This study certainly has merits, possibly even high merits, as a discovery of a species entirely new to science among invasive pest(s) is bordering a sensation (congratulation for the discovery!). But it is so poorly organised and written at this stage, that it is not publishable, and would warrant rejection, not being the material of so high potential interest. 

To beging with, Introduction.

1) Opening the story with global warming is just betraying you of conjuncturalism, but does not make any service in a paper which is about a new species of Sessid moth attacking planted oaks. Global warming might be the reason behind it - but this is interpretation/speculation for Discussion, not for opening sentence. 

2) Second, why not to "introduce" that new species will be described in the paper? Somewhere in the 2nd paragraph - "Why monitoring, we encountered...". Simply write your story. 

3) Third, I wonder why no check-list or monograph of Korean (or nearby countries - China? North Korea? Japan?) Sessidae is not referred to? 

llines 81-2: "is a group of medium to large-sized that resemble" => is a group of medium to large-sized moths that resemble

line 85: "pests" after the second ")"

Methods:

lines 94-9: The sentence does not make sense!  What is "pruning shearsin"

lines 109-114 (crucial, as they describe the sampling protocol for molecular analyses) - again, the description is so confusing that one simple does not know, what was used. More importantly, I cannot see a clear link to the individuals sampled from the oaks.
I recommend starting the whole section with description of obtaining your own sequences, and then, with origins of the database sequences. Further, a rationale for the database sequences is missing, and finally, each species should be referred to by full scientific name, i.e. including the authority and year of secription, at first mention. The latter is of particular importance in taxonomic papers, and your is taxonomic! 

line 116: What exactly is " the head or leg"?

Results
First, I do not really understand the rationale for the detailed diagnosis of Sphecodoptera sheni. It might be warranted (e.g., the ilustrations available in literature are of lower quality / with some traits missing...), but this must be clearly stated somewhere. 

Both the new species diagnosis and the accompanying molecular evidence, on the other hand, seem to be OK. 

Discussion, finally, is very weak, not even trying to address the second aim of your paper from your Introduction. 

Comments on the Quality of English Language

The authors must work with sb. who understands English, as it is not English, but some crazy poorly comprehensible hybrid at this stage. 

Author Response

To Reviewer1

Thank you very much for taking the time to review this manuscript.

We have taken all your comments into consideration and revised the manuscript accordingly. We have marked all changes and corrections with blue in the original manuscript. Point-by-point response to your comments addressed in the attached file. 

Thank you very much again for your valuable comments.

Sincerely, 
Sora Kim

Reviewer 2 Report

Comments and Suggestions for Authors

1.     To facilitate the access to the molecular data used in this study I would strongly recommend authors to deposit their data in BOLD with all metadata for each specimens analysed (georeferenced collecting data,…) and create a data set in BOLD and include its DOI number in the main text. To see how to create a data set and obtain a DOI please check page 26 of the following online manual :

2.     9 sequences from 6 species were taken from BOLD. Please make sure that you have official permission from the project manager where those barcodes are deposited to use them in your manuscript.

3.     Authors should include a table summarizing the specimens used in the molecular systematics analysis, including process IDs, genebank accession numbers, BINs, and the source (paper) from which the barcode was taken. You can follow the supplementary table in the following paper as a model.

2016 Huemer, P., Lopez-Vaamonde, C and Triberti, P. A new genus and species of leaf-mining moth from the French Alps, Mercantouria neli gen. n., sp. n. (Lepidoptera: Gracillariidae). Zookeys 586 : 145-162

4.     Figures 8 & 10 please add to each sample its “sample ID” a unique code that can be found in both BOLD & Genbank

Comments on the Quality of English Language

Needs extensive revision

Author Response

To Reviewer 2,

Thank you very much for taking the time to review this manuscript.

We have taken all your comments into consideration and revised the manuscript accordingly. 

We have marked all changes and corrections with blue in the original manuscript. Point-by-point response to your comments addressed in the attached file. 

Thank you very much again for your valuable comments.

Sincerely, 
Sora Kim

Reviewer 3 Report

Comments and Suggestions for Authors

The work is overall carefully prepared and was a very interesting read for me. I have a few suggestions for minor corrections, but mainly some crucial comments.

There is no reason to believe that Sphecodoptera sheni and Paranthrenella pinoakula are invasive species, which is finally also shown by the presented results of molecular analyses. These are simply two species newly recorded in Korea, which were previously only overlooked, one of them being completely new. Of course, it is possible, though unlikely, that Sphecodoptera sheni has newly spread from China, as the authors state in the discussion on line 329 ("Haplotype analysis suggests that S. sheni may expand its distribution from China"), but this consideration then contradicts their idea that it should be an invasive species. Therefore, I strongly recommend the modification of the title of this paper "Integrated identification and Genetic diversity of the Invasive Clearwing moths (Lepidoptera: Cossidae: Sesiinae) in Korea" to e.g. "Integrated identification and genetic diversity of two Clearwing moths (Lepidoptera: Cossoidea: Sesiidae) recorded in Korea". And in relation to this, then everywhere in the text delete information that these are invasive species. The "fashionable" introduction about climate change (especially the first sentence, similarly the useless citations 1 and 2) and about the introduction of non-native species is irrelevant, but so be it.

Paranthrenella pinoakula and Sphecodoptera sheni are not related species (each belongs to a different tribe of the subfamily Sesiinae), why are molecular genetic results of these two species compared and presented together (e.g. in figs 8, 9, 10)? Sphecodoptera sheni is not compared with related species, so only the results of the molecular analysis of species of the genus Paranthrenella are interesting and useful.

Paranthrenella pinoakula is described by only 3 males, and the female is even not known. How is this possible when the authors consider the species a pest?

L. 81 Lepidoptera: Cossidae: Sesiinae, should be Lepidoptera: Cossoidea: Sesiidae

L. 147 Sesia Spatenka, Lastuvka, Gorbunov, Tosevski & Arita, 1993: 87 – Spatenka, Lastuvka, Gorbunov, Tosevski & Arita, 1993 are not the authors of the name Sesia; it should be written differently (e.g. Sesia, sensu ... or Sesia; ...) and the publication is not listed in the references.

L. 154 Sphecodoptera sheni Kallies, Arita, Owada & Wang, 2014: 587 – Kallies, Arita, Owada & Wang, 2014 are not the authors of the name Sphecodoptera; again, the name must be written differently and the publication added to the references.

L. 213 vernation, antermedially – venation, anteromedially (or antemedially)

L. 225 Asymmetrical valvae are very remarkable. How many individuals did you study? Is this not teratology?

L. 242 Supplement the authors of names here or elsewhere at the appropriate place, this applies to all species named in the text.

You should somewhere (e.g. in the remark at the end of the description) state how many species of the genus Paranthrenella are known and from where. Important papers that (also) deal with species of this genus are not cited, in particular

Kallies A. 2020: Zootaxa 4833 (1): 001–064

Tiantian Yu. et al. 2021: Zootaxa 4920 (1): 123–130

Author Response

To Reviewer 3,

Thank you very much for taking the time to review this manuscript.

We have taken all your comments into consideration and revised the manuscript accordingly. 

We have marked all changes and corrections with blue in the original manuscript. Point-by-point response to your comments addressed in the attached file. 

Thank you very much again for your valuable comments.

Sincerely, 
Sora Kim

Round 2

Reviewer 1 Report

Comments and Suggestions for Authors

The manuscript has much improved. Good job!

Reviewer 3 Report

Comments and Suggestions for Authors

I have no further comments. I see that you continue to regard the species found as invasive or potentially invasive, for which I see no reason, but that is your position.